# Axitinib Rechallenge Restores the Anticancer Effect after Nivolumab: A Case Report

**DOI:** 10.3390/ijms241512149

**Published:** 2023-07-29

**Authors:** Yueh-Shih Chang, Pei-Hung Chang, Deng-Huang Wang, Chun-Bing Chen, Chi-Ying F. Huang

**Affiliations:** 1Hemato-Oncology, Department of Internal Medicine, Chang Gung Memorial Hospital, College of Medicine, Keelung & Chang Gung University, Taoyuan 33302, Taiwan; 2Institute of Clinical Medicine, National Yang Ming Chiao Tung University, Taipei 112304, Taiwan; 3Institute of Biopharmaceutical Sciences, National Yang Ming Chiao Tung University, Taipei 112304, Taiwan; 4Department of Dermatology, Drug Hypersensitivity Clinical and Research Center, Chang Gung Memorial Hospital, Taipei, Linkou, Keelung 833301, Taiwan; 5Taiwan Graduate Institute of Clinical Medical Sciences, College of Medicine, Chang Gung University, Taoyuan 330036, Taiwan; 6Immune-Oncology Center of Excellence, Chang Gung Memorial Hospital, Linkou, Taoyuan 333, Taiwan

**Keywords:** renal cell carcinoma, nivolumab, axitinib, PD-1, cytotoxic

## Abstract

The immune checkpoint inhibitor/tyrosine kinase inhibitor (ICI/TKI) combination treatment is currently the first-line treatment for metastatic renal cell carcinoma (mRCC). However, its efficacy beyond the third-line setting is expected to be relatively poor, and high-grade toxicities can develop by prior exposure to multiple drugs, resulting in a relatively poor performance in patients. Determining the best treatment regimen and sequence remains difficult and requires further investigation in patients with mRCC. In this study, two cases of mRCC, who failed several lines of TKI and nivolumab but exhibited a good anticancer effect after rechallenging with axitinib, are described. Both patients had a faster time to best response and better progression-free survival (PFS) than during previous treatments. Moreover, the axitinib dose could be reduced to 2.5 mg daily when used in combination with nivolumab while continuing to exert an impressive anticancer effect. To determine the cytotoxic effect, we performed a lymphocyte activation test and found that the level of granzyme B released by cytotoxic T lymphocytes and natural killer cells was higher when axitinib was combined with nivolumab. To evaluate this result, a bioinformatics approach was used to analyze the PRISM database. In conclusion, based on the results of a lymphocyte activation test and PD-1 expression, our findings indicate that sequential therapy with axitinib rechallenge after nivolumab resistance is reasonable for the treatment of mRCC.

## 1. Introduction

At present, treatment for mRCC in patients after the failure of immune checkpoint inhibitor treatment is lacking. However, in the National Comprehensive Cancer Network Guidelines for mRCC, treatment options for sequential therapy are available, including other ICI, tyrosine kinase inhibitors (TKIs), mammalian target of rapamycin (mTOR) inhibitors, and bevacizumab [1]. In this context, no standard guidelines for the sequential therapy of mRCC have been established. Although the ICI/TKI combination treatment shows an excellent overall response rate (ORR), PFS, and overall survival (OS), the efficacy of combinations is influenced by prior treatments. For example, the first-line ORR in the KEYNOTE-426 and Javelin renal 101 trials was 52.5–59.3% [2,3]. Similarly, in two other early-phase clinical trials, an impressive ORR of 55.8–62% was obtained for the ICI/TKI combination treatment in ICI-pretreated patients [4,5]. Therefore, combination treatment has been established as a first-line treatment for mRCC, as well as a second-line treatment for patients with refractory ICI. Currently, clinical trial data does not provide clinicians with information regarding optimal drug sequencing and selection beyond the second-line setting. In practice, the ORR for the fourth-line ICI/TKI utilization is low, at only 21% [5]. As ICIs can affect patient response to subsequent cytotoxic agents, determining the best sequence remains a challenge, and further investigation is needed. When nivolumab is applied as a third-line treatment, it is crucial for physicians to determine the most effective sequential treatment after nivolumab fails. In this study, we report on two cases of heavily treated mRCC in which axitinib restored drug sensitivity after nivolumab treatment. In this context, the PRISM database was used to understand the effect of programmed death-1 (PD-1) expression on axitinib to determine the best line of treatment for patients with mRCC showing nivolumab resistance.

## 2. Case Description

Case 1: A 63-year-old man with exertional dyspnea visited our hospital in April 2018. In the past, he had been in good health without any chronic disease. Whole-body computed tomography (CT) revealed a mass of nearly 10 cm in the kidney and lung consolidation, both on the left side of the corresponding organs. A CT-guided lung biopsy was performed, and stage IV renal clear cell carcinoma with lung metastases was diagnosed (Figure 1A, left side). The patient had an intermediate risk score and was initiated with a course of pazopanib as the first-line treatment. Only a partial response was achieved during pazopanib treatment, according to the response evaluation criteria in solid tumors. A left nephrectomy was performed. After 10 months, he suffered from shortness of breath again, and upon examination, the lung lesion was found to have progressed, as confirmed by CT in Figure 1B. There was no local recurrence at his primary cancer site after nephrectomy in Figure 1B(i–iii). Thus, his treatment response to sequential therapies was poor. Subsequently, everolimus was chosen as the second-line treatment. However, his clinical condition failed to improve after one month. Next, axitinib was chosen as the third-line treatment, for which a duration of response of only two months was achieved (Figure 1B(iv) and Appendix A). Moreover, the patient suffered from several adverse effects resulting from axitinib treatment (10 mg daily), including grade 3 hypertension and grade 2 decreased appetite, according to the Common Terminology Criteria for Adverse Events. After rapid failure to respond to the three types of TKI, the patient was initiated nivolumab for fourth-line treatment. However, the disease continued to progress after six months (Figure 1B(v)). Pseudoprogression had been taken into consideration, but in the last month of nivolumab monotherapy, the patient became completely bedridden due to near respiratory failure. Lung biopsy or broncho lavage was difficult to perform to exclude pseudoprogression. Because of drug availability, we reinstated axitinib (10 mg daily) with nivolumab (200 mg every two weeks). His clinical symptoms and performance status improved markedly within one week (Appendix A). Based on his history of adverse events, the dosage of axitinib was decreased to 5 mg/day after one month. After three months, the patient was presented with one small residual lung lesion (Figure 1B(vi)), based on CT observation, but continued to experience grade 3 adverse effects due to axitinib. The dose of axitinib was reduced to 2.5 mg/day alongside the standard dose of nivolumab. To understand whether a lower dose of axitinib influenced the efficacy of the ICI/TKI combination treatment, a lymphocyte activation test was used to analyze the cytotoxic effect (Table 1). The patient’s clinical condition and disease remained under control for two years, and the dosage of axitinib was maintained at 2.5 mg/day. The clinical course and laboratory findings are shown in Appendix A. In previous clinical trials for ICIs, nivolumab use was two years. Nivolumab was discontinued for six months, and the patient’s status remained fair. To date, our patient is only receiving axitinib at a dose of 2.5 mg daily, with recent CT results confirming his stable disease (Appendix A). Case 2: A 62-year-old man with gross hematuria visited our hospital in 2012. The CT results revealed a mass of 5 cm on the right side of the kidney. As such, a radical nephrectomy was performed. His final diagnosis was stage IIIa clear cell renal cell carcinoma (shown in Figure 1A, right). While the benefits of adjuvant TKI at that time were unclear, his disease recurred soon in 2013. His recent CT findings indicated multiple lung and pleural metastases (Figure 1C). His primary site has been clear since his operation in 2012 (Figure 1C(I–III)). Unlike our first patient, this patient’s disease showed a good response to each line of treatment. The patient was administered pazopanib as the first-line treatment for nearly five years. When the disease progressed in 2018, targeted therapy was shifted to everolimus as the second-line treatment. After seven months, axitinib was administered as the third-line therapy for an additional seven months (Figure 1B(iv)). Next, monotherapy with nivolumab was administered as the fourth-line treatment, with the patient achieving a partial response (Figure 1B(v)). Although the effect of nivolumab lasted for 18 months, multiple lung nodules with pleural and rib invasion were subsequently observed. Re-biopsy after nivolumab administration indicated that the pathology was clear cell carcinoma. Following this, axitinib therapy was reinstated as monotherapy, with the time to best response being one week. Nearly no residual tumor was observed after nine months of treatment with axitinib (Figure 1B(vi) and Appendix A), which differed from this patient’s prior experience with axitinib. His clinical course, laboratory findings, and current image are shown in Appendix A.

## 3. Discussion

Variable sequential treatments for metastatic clear cell renal cell carcinoma have been used in the clinical setting, including mTOR inhibitors, another VEGF-TKI, and immunotherapy. The two cases showed a marked and rapid response when axitinib was rechallenged after nivolumab. In the present study, one patient was administered a combination treatment of nivolumab and axitinib, while another patient received axitinib only. Both patients achieved a rapid and better response within one week compared with their first-time response to axitinib. The efficacy of the axitinib sequential use and rechallenge is summarized in Table 2. In a small retrospective study, the median PFS after the axitinib rechallenge was only 3.3 months, with a median OS of 21.8 months [6]. Although other studies have also reported on the efficacy of rechallenging axitinib, the timing of axitinib varied depending on the physician’s choice [6,7,8]. Among those patients who received axitinib rechallenge after ICIs, half showed a partial response. Among the cases reported in the present study, the disease burden decreased rapidly, and a nearly complete response was observed in Case 2 (axitinib monotherapy). Response to prior treatment with TKI or ICI could not predict the effect of rechallenging axitinib. Currently, in the BIONIKK trial, it has been shown that choosing the first-line treatment based on the tumor molecular phenotype of metastatic clear cell renal cell carcinoma is feasible and efficacious [9]. Therefore, bioinformatics analysis and a lymphocyte activation test were performed after treatment for further investigation. To understand how axitinib regains drug sensitivity in rechallenged patients, we analyzed the associations between axitinib and PD-1 expression using a large data platform according to systematic viability profiling (Figure 2A). PRISM is a database of high-throughput experiments based on gene barcode-tagged mixture culture systems [10,11]. Using the PRISM database, the association between the cell viability of RCC cell lines under different messenger ribonucleic acid (mRNA) expressions of programmed cell death protein 1 (PD-1) was analyzed using Pearson’s correlation analysis (Figure 2A). As a result, a negative correlation was observed between the axitinib sensitivity score and PD-1 expression in RCC cell lines, indicating that the RCC cell lines with high PD-1 expression were significantly sensitive to axitinib (*p* = 0.04; Figure 2B, left). Unlike PD-1, no correlation was observed between programmed death-ligand 1(PD-L1) and the RCC cell line (Appendix A). To understand the change in PD-1 expression, PD-1 expression was analyzed by immunohistochemical staining (Figure 3) in Case 2. As a result, PD-1 expression after nivolumab treatment was found to be slightly higher than before. This finding is consistent with our results from the PRISM database but different from previous findings of PD-1 receptor internalization after nivolumab treatment. The impact of the overexpression of PD-1 on RCC still needs further investigation [12,13]. Previous studies have focused on programmed death-ligand 1 (PD-L1), with high levels of PD-L1 expression being reported in the high-risk RCC group [14]. Unlike lung cancer and melanoma, the expression of PD-L1 in RCC does not provide better treatment efficacy when using ICIs. Therefore, further research is needed to confirm our findings. Another crucial issue is whether to choose combination therapy or TKI monotherapy. Regarding resistance to anti-PD-1/PD-L1 therapy, the influence of the immunosuppressive tumor microenvironment (TME) is an important factor [13,15], which can be adjusted by ICIs to restore drug sensitivity. Many other factors can influence the choice of sequential therapy, including the associated financial burden. Since most studies on combination treatment were designed in the context of first-line treatment, the ideal duration for nivolumab in the fourth-line setting is not known. With regard to Case 1, this patient was only administered nivolumab for two years, according to the National Health Insurance. After the discontinuation of nivolumab, this patient was then administered axitinib only until the present day. As a result, his cancer behavior has changed, becoming relatively indolent or growing slowly. Therefore, for patients who have never received ICI/TKI combination treatment, it remains worth considering this combination therapy. In addition, in Taiwan, nivolumab alone is approved for mRCC after the failure of at least two TKIs or mammalian targets of rapamycin (mTOR) inhibitors. Moreover, after the use of nivolumab, sequential drugs are not reimbursed by the national insurance of Taiwan. Therefore, physicians must determine the most effective sequential treatment. The clinical insights reported in this study provide information that will be helpful to physicians making these decisions. The third-line treatment was axitinib when used in combination with other drugs. We hypothesized that axitinib could enhance the anticancer effects of other drugs via an increased expression of the tumor cell-intrinsic PD-1 receptor as a means to overcome resistance to PI-1 blockade therapy [16]. Meanwhile, the effects of nivolumab on antibody-dependent cellular cytotoxicity persisted. To demonstrate this, we performed a lymphocyte activation test to measure granzyme B released from the memory T cell to nivolumab in vitro (Table 1) [17]. This test was originally performed in 2019 to confirm the immune-related adverse events in Case 2. Because the immune-related adverse effect is also a potential predictor of a patient’s response to immunotherapy (13), this test was performed for a second time in 2020 to determine whether nivolumab had failed. The level of granzyme B was found to be higher when the drugs were combined (axitinib 2.5 mg/day and nivolumab 200 mg every two weeks). Similar results have been reported previously, indicating that fewer regulatory T cells were associated with tumor regression [18] when a relatively low dose of axitinib was combined with nivolumab. Therefore, the combination of a low dose of axitinib and nivolumab shows potential as a sequential treatment for mRCC, as well as being better than the standard dose of axitinib because it results in less toxicity and shows better efficacy.

## 4. Conclusions

In conclusion, although the optimal sequence of drug administration for mRCC has yet to be determined, in this study, rechallenging with axitinib after nivolumab refractory disease treatment was found to achieve a better response compared to its first-time use alone. Furthermore, even after reducing the dose of axitinib, its anticancer effect remained satisfactory when combined with nivolumab. Two potential biomarkers using a lymphocyte activation test or PD-1 IHC staining could help determine the drug effect. In the future, the cell viability, cytotoxicity assessment, and real-world evidence study will be conducted to gather more evidence of the rechallenge of axitinib after nivolumab.

## 5. Method

### 5.1. PRISM Method

This case report uses a PRISM bioinformatics system to screen cancer cell line mixtures labeled with 24-nucleotide barcodes [10,11]. The mixtures are placed into tissue culture assay plates and treated with the targeted compounds and vehicles (DMSO) controls. Genomic DNA is obtained from the residual viable cells from the mixtures. The polymerase chain reaction amplifies the barcode sequences. The barcode sequences are added to individual microbeads with antisense barcode sequences and to streptavidin-phycoerythrin. The fluorescent signal from each bead is detected by a Luminex FlexMap detector. For adjusting different barcode efficiencies and cell doubling times, the signal from each cell line is placed to that of vehicle-treated control. Based on the methods, it can reveal the inhibition profiles for a targeted drug through multiple cell lines. The statistically significant result of each gene level was using the MAGeCK-MLE method, and two-sided *p*-values were corrected for multiple hypothesis testing by the Benjamini–Hochberg method.

### 5.2. Lymphocyte Activation Test

The lymphocyte activation test detects the memory T cells stimulated by a specific drug. The peripheral blood mononuclear cells are co-incubated with the specific drug in vitro, and then Granzyme B is measured by Coomassie blue protein assay (Bio-Rad), ELISA Western blotting, and Coomassie staining of 17% SDS-PAGE gels [17]. Granzyme B is one of the serine proteases granzymes released by activated T cells and is used to determine drug hypersensitivity. The higher granzyme B represents the more cytotoxic T cell activation. And two reagents (PBS and PHA) are used as control groups to stimulate T cells to compare with the IO drugs.

## Figures and Tables

**Figure 1 ijms-24-12149-f001:**
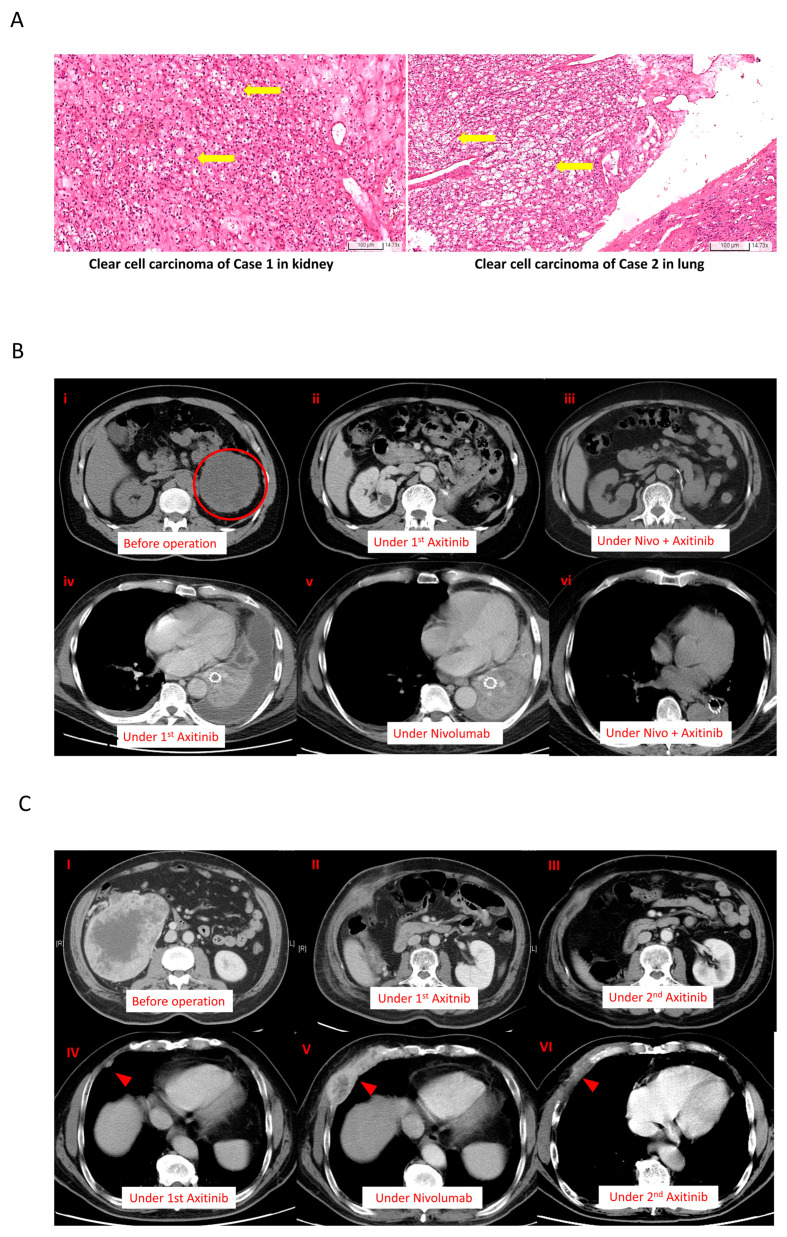
The clinical course and treatment response of patients according to the main tumor lesion. (**A**) The cells were presented with clear cytoplasm and arranged in nests in both cases (yellow arrows); (**B**) cancer response and treatment duration in Case 1. (i–iii) The primary cancer site of Case 1; there was no cancer recurrence until now. (iv) The best response of treatment with first-time axitinib. (v) The response of treatment with nivolumab. (vi) After nivolumab treatment, the cancer cells became more sensitive to second-time axitinib. (**C**) Similar cancer responses were observed in Case 2 via both chest radiographs and CT. (I–III) After the nephrectomy, the primary site has been in complete remission until now. (IV) The response of treatment with first-time axitinib. (V) The response of treatment with nivolumab. (VI) After nivolumab treatment, the tumor shrank markedly in size during the second-time treatment with axitinib.

**Figure 2 ijms-24-12149-f002:**
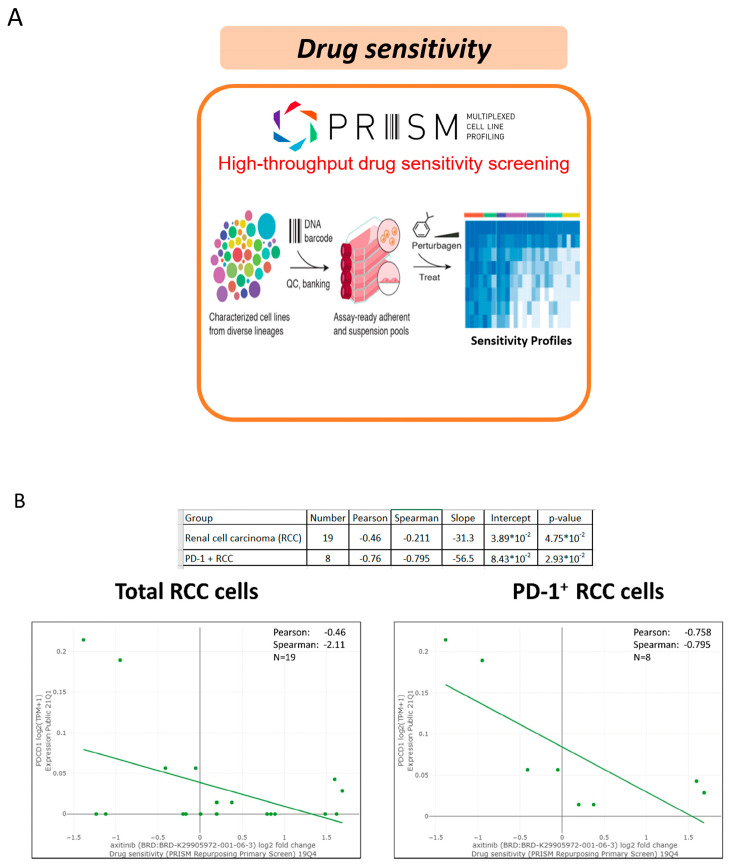
The association of the tumor-intrinsic programmed cell death protein 1 (PD-1) in the RCC cell line via the PRISM database and drug sensitivity to axitinib. (**A**) The PRISM database facilities drug screening; (**B**) Drug sensitivity is associated with the status of PD-1 expression.

**Figure 3 ijms-24-12149-f003:**
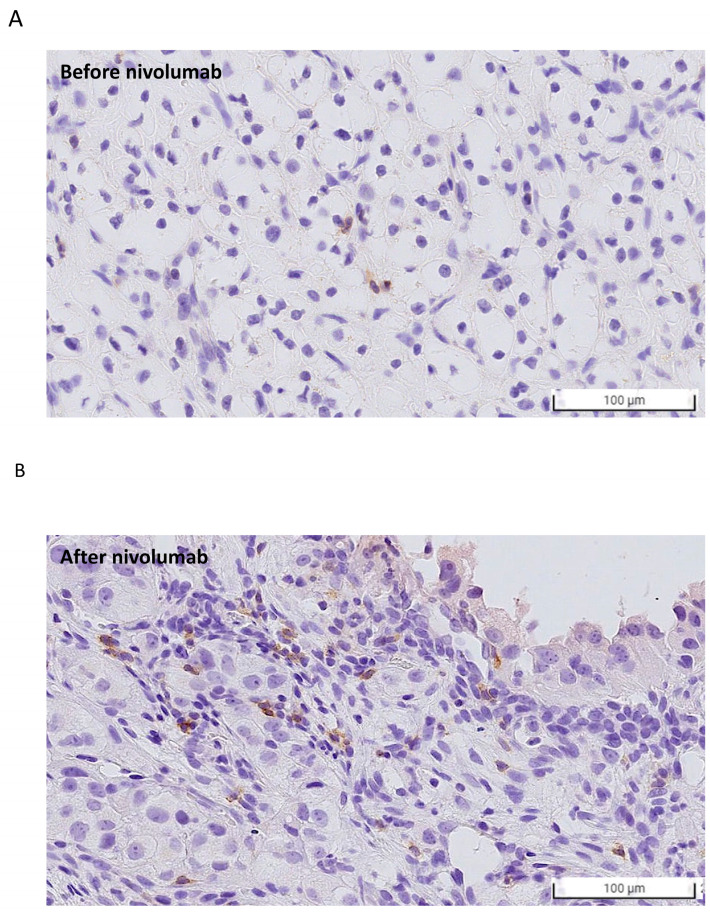
PD-1 expression evaluated using immunohistochemistry (IHC) staining (**A**) before and (**B**) after nivolumab treatment. The number of PD-1-positive cancer cells increased after nivolumab treatment, which is in line with the findings based on the PRISM database.

**Table 1 ijms-24-12149-t001:** Clinical changes in the lymphocyte activation test. A lymphocyte activation test showed a higher level of granzyme B release after the axitinib rechallenge.

**20190829 (Nivolumab Only, After Starting Treatment One Month)**		
	Granzyme B	
Drug	(pg/mL)	fold
PBS	203.0	1.00
Nivolumab	408.8	2.01
Ipilimumab	219.2	1.08
Pembrolizumab	554.0	2.73
PHA	11,686.4	57.57
**20200313 (Nivolumab and 2.5 mg Axitinib)**		
PBS	102.5	1.00
Nivolumab	362.8	3.54 *
PHA	20,368.0	198.79

Abbreviation: PHA: mitogen phytohemagglutinin, PHS: dissolved in phosphate-buffered saline.

**Table 2 ijms-24-12149-t002:** The literature review about the sequential therapy and rechallenge of axitinib.

Study Type	Title	Efficacy	Compared with Our Cases	Reference
Case series	Efficacy of Axitinib After Nivolumab Failure in Metastatic Renal Cell Carcinoma (2020)	Efficacy of axitinib as a third-line therapy after the failure of a first-line VEGFR-TKI and a second-line nivolumab monotherapy for mRCC. The median PFS was 12.8 months, and the 1-year and OS rate was 71.6%	Similarities: The efficacy of axitinib after nivolumab would be better Differences: We focus on the efficacy of axitinib rechallenge after nivolumab	[6]
Case series	Efficacy of axitinib rechallenge in metastatic renal cell carcinoma (2021)	PFS in axitinib rechallenge: 3.3 months (95% CI 6.9–not reached_ OS in axitinib rechallenge: 21.8 months (95% CI 6.9–not reached	Similarities: The efficacy of the axitinib rechallenge was revealed in this study. The PFS is only 3.3 months Differences: We focus on the specific timing (after nivolumab)	[7]
Case report	Axitinib Reverses Resistance to Anti-Programmed Cell Death-1 Therapy in a Patient With Renal Cell Carcinoma (2022)	Axitinib successfully reversed primary resistance to anti-PD-1 therapy in a patient with RCC	Similarities: Axinitib can reverse the primary opposition to anti-PD-1 treatment Differences: In this study, anti-PD-1 treatment is the first line setting.	[8]

Abbreviation RCC = renal cell carcinoma; VEGFR-TKI= vascular endothelial growth factor receptor tyrosine kinase inhibitor; PFS= progression-free survival; OS= overall survival; anti-PD-1= Anti programmed cell death protein-1. Table legend: Three critical articles are associated with axitinib’s sequential therapy and rechallenge. According to a previous study, the median progression-free survival of axitinib rechallenge is only 3.3 months. Therefore, how to rechallenge axitinib is a critical issue.

## Data Availability

The original data generated and analyzed for this study are included in the published article. The corresponding author can be contacted for additional requirements.

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
