# Peer review of "Axitinib Rechallenge Restores the Anticancer Effect after Nivolumab: A Case Report"

_ijms, 2023, doi:10.3390/ijms241512149_

Round 1

Reviewer 1 Report (Previous Reviewer 3)

This manuscript reports on two cases of metastatic renal cell carcinoma treated with axitinib as late line therapy, which showed a certain level of response. 

Since there is no consistent method for late line treatment of metastatic renal cell carcinoma, I think that this report will be helpful in actual clinical practice. 

However, the first case was treated with the combination of axitinib and nivolumab in the late line, and was not a pure axitinib rechallenge. 

It is not clear if the combination of IO/TKI in the late line is available worldwide. It is NOT possible, at least not in the reviewer's country. 

Also, the patient uses 2.5 mg of axitinib, but as axitinib is available in 1 mg and 5 mg formulations, it is not clear how the 2.5 mg dose was administered. 

It is also unclear how much the results of this study will be reflected now that IO-Combo and IO/TKI are commonly used as the first line. 

As a paper, it is more like a case report than an original article, and the overall text is too long. Supplementary Figute1B is unnecessary, and Figures 1(b)(C), 1C, and 1D should be summarized. 

Please explain how we should interpret the results of Lymphocyte activation test in Figure 2, Supplementary Figure 2 and Table 1. Please explain how GranzymeB is generally used and how it relates to the results of this study. 

In the Case presentation, it is mentioned twice regarding Case 2. 

There is a spelling error: 5. Methods →Methods? 

There is no explanation of abbreviations in Table 1 (PBS, PHA). 

Specific dates in Figure and Table should be removed. 

Figure should be improved as the overall image quality is poor. 

Author Response

Response to reviewer 1:

Comment 1:

This manuscript reports on two cases of metastatic renal cell carcinoma treated with axitinib as late line therapy, which showed a certain level of response. Since there is no consistent method for late line treatment of metastatic renal cell carcinoma, I think that this report will be helpful in actual clinical practice. However, the first case was treated with the combination of axitinib and nivolumab in the late line, and was not a pure axitinib rechallenge. It is not clear if the combination of IO/TKI in the late line is available worldwide. It is NOT possible, at least not in the reviewer's country. Also, the patient uses 2.5 mg of axitinib, but as axitinib is available in 1 mg and 5 mg formulations, it is not clear how the 2.5 mg dose was administered. It is also unclear how much the results of this study will be reflected now that IO-Combo and IO/TKI are commonly used as the first line.

Response:

Thank you for your comprehensive suggestion. Our case report provides two possible treatments of axitinib rechallenge after nivolumab failure, including TKI monotherapy and IO/TKI combination. In our country, the combination of IO/TIK is not reimbursed by our National Health Care System. Therefore, our case 1 must pay by himself for the combination IO/TIK. According to the clinical response of case 1, the reduced dose of axitinib was administered only when combined with nivolumab. Another case report (reference 8) supports our findings. In this case report, the dose of axitinib also needs reduction. For case 2, axitinib was reinstated as monotherapy in a standard dose.

Comment 2:

As a paper, it is more like a case report than an original article, and the overall text is too long. Supplementary Figute1B is unnecessary, and Figures 1(b)(C), 1C, and 1D should be summarized. Please explain how we should interpret the results of Lymphocyte activation test in Figure 2, Supplementary Figure 2 and Table 1. Please explain how Granzyme B is generally used and how it relates to the results of this study

Response:

We appreciate your suggestion. We have adjusted our text and figures accordingly. The result of the lymphocyte activation test and granzyme B have been highlighted in our manuscript. Granzyme B is one of the serine protease granzymes released by activated T cells and is used to determine drug hypersensitivity in our cases. Our past studies also found a potential association between the iRAE and IO treatment efficacy. And we used two reagents (PBS and PHA) to stimulate T cells to compare with the IO drugs.

In the text of Methods:

Granzyme B is one of the serine protease granzymes released by activated T cells and is used to determine drug hypersensitivity. And two reagents (PBS and PHA) are used as control groups to stimulate T cells to compare with the IO drugs.

Comment 3:

In the Case presentation, it is mentioned twice regarding Case 2. There is a spelling error: 5. Methods →Methods? There is no explanation of abbreviations in Table 1 (PBS, PHA). Specific dates in Figure and Table should be removed. Figure should be improved as the overall image quality is poor.

Response:

Thank you for your comprehensive suggestion. We check the number of Case 2 and ensure only one is mentioned about Case 2. The spelling error of Methods has been adjusted. Initially, the abbreviations of PBS and PHA were mentioned at the end of figure legends (mitogen phytohemagglutinin (PHA) and dissolved in phosphate-buffered saline (PBS)). We also highlighted the abbreviations in the text of methods. The date in Figures and Tables also has been removed. We also improved the quality of the Figures.

Reviewer 2 Report (New Reviewer)

1. It is good to label the histology picture of the kidney

2. "Variable sequential treatments for metastatic clear cell renal cell carcinoma have been 167 used in the clinical setting." Please name some of the sequential treatments here.

3. Please explain what are statistical analysis used by the author in this study in the methods section. 

4. Please provide the suggestion for future studies in the conclusion section

Overall good.

Author Response

Response to Reviewer 2:

Comment 1:

It is good to label the histology picture of the kidney

Response:

Thank you for your suggestion. We provide the labeled histology pictures in the revised Figure 1A. In the revised Figure 1A, we labeled and added a description “ the cells with clear cytoplasm, arranged in nests in both cases.”

Comment 2:

"Variable sequential treatments for metastatic clear cell renal cell carcinoma have been 167 used in the clinical setting." Please name some of the sequential treatments here.

Response:

According to the current guideline, we added the sequential metastatic clear cell carcinoma treatment in this sentence. And the revised one is “Variable sequential treatments for metastatic clear cell renal cell carcinoma have been used in the clinical setting, including mTOR inhibitor, another VEGF-TKI, and immunotherapy.”

Comment 3:

Please explain what are statistical analysis used by the author in this study in the methods section. 

Response:

We appreciate your suggestion and mention the statistical analysis in the methods section. “The statistically significant result of each gene level was using the MAGeCK-MLE method and two-sided p-values were corrected for multiple hypothesis testing by the Benjamini-Hochberg method.”

Comment 4:

Please provide the suggestion for future studies in the conclusion section

Response:

Thank you for your valuable suggestion. Thus, we adjusted our conclusion and added “In the future, the cell viability, cytotoxicity assessment, and real-world evidence study will be conducted to gather more evidence of re-challenge of axitinib after nivolumab.

This manuscript is a resubmission of an earlier submission. The following is a list of the peer review reports and author responses from that submission.

Round 1

Reviewer 1 Report

The authors report that axitinib rechallenge restores the anti-cancer effect after nivolumab. The clinical courses of patients are interesting. However, biomarker analysis is too poor to explain anticancer effects. In the IHC analysis, it is very difficult to objectively understand that PD-1 expression level increases after nivolumab treatment. Moreover, there is no results of a lymphocyte activation test (Table1 is missing). Even if Table 1 were included, this biomarker analysis would not explain the anti-tumor effect of the axitinib challenge.

There are many other shortcomings in terms of the paper's presentation. It is extremely difficult to understand the clinical courses of two patients due to the inappropriate representation of CT images. Furthermore, the paper does not maintain the appearance of a paper because the results of the biomarker analysis are listed in the discussion and there is no section on methods.

Author Response

Response to reviewer 1
Comments and Suggestions for Authors:
Comment 1:
The authors report that axitinib rechallenge restores the anti-cancer effect after nivolumab. The clinical courses of patients are interesting. However, biomarker analysis is too poor to explain anticancer effects. In the IHC analysis, it is very difficult to objectively understand that PD-1 expression level increases after nivolumab treatment. Moreover, there are no results of a lymphocyte activation test (Table 1 is missing). Even if Table 1 were included, this biomarker analysis would not explain the anti-tumor effect of the axitinib challenge.
Response:
We appreciate your comments and suggestion. Indeed, the expression of PD-1 is an incidental finding, and we believe further investigation is warranted. Therefore, we summarized previous studies about axitinib rechallenge in Table 2. We also connected with other authors who reported the axitinib rechallenge's efficacy and hoped they could perform an IHC analysis. But we have yet to receive an answer. Therefore, in the future, we will gather more patients in our hospital to understand the mechanism. We added Table 1 to the manuscript, and the timeline is highlighted. According to a previous study, the PFS of axitinib rechallenge is only 3.3 months. Thus, it is crucial to understand which kind of sequential use provides maximal clinical benefit. Our biomarker analysis shows that the drug sensitivity of axitinib is associated with PD-1 expression. Our finding is premature but could provide us with a possible biomarker.
Comment 2:
There are many other shortcomings in terms of the paper's presentation. It is extremely difficult to understand the clinical courses of two patients due to the inappropriate representation of CT images. Furthermore, the report does not maintain the appearance of a paper because the results of the biomarker analysis are listed in the discussion and there needs to be a section on methods.
Response:
Thank you for your critical suggestion. Based on your advice, we added the pre- and post-axitinib CT images in supplementary figures 1C and 1D. And we add a section on methods for biomarker analysis and lymphocyte activation tests. Thank you for your valuable advice.
Sincerely yours,
Yuehshih Chang

Reviewer 2 Report

The authors describe 2 cases of RCC that responded to axitinib rechallenge. 

1) Case 1 :Multiple therapies were given. It is important to show/describe imaging results pre and post axitinib  and then pre and post axitinib rechallenge. Also the patients disease bulk needs to be clarified. It does not appear that case 1 has high volume pulmonary disease.

2) Case 1 - Pseudoprogression needs to be considered after nivolumab showing progression followed by response. Even though nivolumab was stopped, its long half life has to be taken into consideration. These possibilities need to be added to the discussion of case 1

3) Please discuss literature reports of VEGF TKI rechallenge and compare this case to reports already published.

4) Case 2 : need the scans pre and post axitinib the first time and after rechallenge. 

5) In the US axitinib and pembrolizumab is standard frontline therapy so synergy has been established.

6) The LAK cell assay should be described and the timeline represented clearly. 

Author Response

Response to reviewer 2

Comments and Suggestions for Authors:

Response:

We appreciate your comments. We have revised our manuscript accordingly. Thank you for your valuable advice.

Comment 1:

  • Case 1: Multiple therapies were given. It is important to show/describe imaging results pre and post axitinib and then pre and post axitinib rechallenge. Also the patients disease bulk needs to be clarified. It does not appear that case 1 has high-volume pulmonary disease.

Response:

Thank you for your suggestion. The new imaging results about pre- and post-axitinib are provided in the supplementary figures 1C and 1D and showed high volume pulmonary metastasis in CXR and CT images.

Comment 2

  • Case 1 - Pseudoprogression must be considered after nivolumab showing progression followed by the response. Even though nivolumab was stopped, its long half-life must be considered. These possibilities need to be added to the discussion of case 1

Response:

Thank you for your reminder. Pseudoprogression is a critical issue for ICIs treatment because it causes early discontinuation of therapy due to the false judgment of progression. According to the clinical course of case 1, the progression time is six months after nivolumab treatment, and the patient’s performance status deteriorated rapidly. Due to nearly respiratory failure, lung biopsy or broncho lavage was difficult to perform to exclude pseudoprogression. And CT image in Figure 1B showed cancer enlargement. We also prescribed high-dose steroids and antibiotics, but his clinical condition did not improve. However, his clinical symptoms and X-ray improved dramatically after adding axitinib, which did not mention in other studies or case reports (supplementary figure 2C). In the section case description, we mentioned “Pseudoprogression had been taken into consideration, But in the last month of nivolumab monotherapy, the patient became completely bedridden due to near respiratory failure. Lung biopsy or broncho lavage was difficult to preformed to exclude pseudoprogression”

Comment 3

  • Please discuss literature reports of the VEGF TKI rechallenge and compare this case to published reports.

Response:

Thank you for your suggestion. We did the literature review about the sequential use and rechallenge of axitinib in Table 2. At the same time, the differences were also pointed out.

Comment 4:

4) Case 2: need the scans pre and post-axitinib the first time and after rechallenge. 

Response:

Based on your advice, we added the pre and post axitinib CT images in the supplementary figure 1C.

Comment 5:

  • In the US, axitinib and pembrolizumab are standard frontline therapy so synergy has been established.

Response:

Indeed, the efficacy of the combination of axitinib and pembrolizumab in 1st line setting is very impressive, so the combination has become one of the standard frontline therapies. However, both our cases received axitinib and nivolumab after third-line treatment. In our case report, we focus on the efficacy of axitinib rechallenge and a possible treatment choice when physicians need to choose sequential VEGF TKI.

Comment 6:

6) The LAK cell assay should be described, and the timeline should be represented clearly. 

Response:

Thank you for your reminder. Because our article type is the case report, we added the method of LAK cell assay in the discussion (we performed a lymphocyte activation test to measure granzyme B released from the memory T cell to nivolumab in vitro) and the method. At the same time, the timeline is highlighted in table 1

Sincerely yours,

Yuehshih Chang

Reviewer 3 Report

Interesting paper on axitinib rechallenge for mRCC in real practice. The reviewer also had experience with cases in which axitinib rechallenge was successful. I think this is an important finding in the ICI era.

Regarding Case 1, strictly speaking, it is a combination of nivolumab and axitinib, but has this been reported before? Has this combination been covered by insurance?

In Figure 1B, there is an image of what appears to be a stent in the lung lesion. There is no mention of when it was used.

Author Response

Response to reviewer 3
Comments and Suggestions for Authors: The immune checkpoint inhibitor/tyrosine kinase inhibitor (ICI/TKI) combination treatment is currently the first-line treatment for metastatic renal cell carcinoma (mRCC). However, its efficacy beyond the third-line setting is expected to be relatively poor, and high-grade toxicities can develop by prior exposure to multiple drugs, resulting in a relatively poor performance in patients. Determining the best treatment regimen and sequence remains difficult and requires further investigation in patients with mRCC. In this study, two cases of mRCC, who failed several lines of TKI and nivolumab but exhibited a good anti-cancer effect after rechallenge with axitinib, are described. Both patients had a faster time to best response and better progression-free survival (PFS) than during previous treatments. Moreover, the axitinib dose could be reduced to 2.5 mg daily when used in combination with nivolumab while continuing to exert an impressive anti-cancer effect. To determine the cytotoxic effect, we performed a lymphocyte activation test and found that the level of granzyme B released by cytotoxic T lymphocytes and natural killer cells was higher when axitinib was combined with nivolumab. To evaluate this result, a bioinformatics approach was used to analyze the PRISM database. In conclusion, based on the results of a lymphocyte activation test and PD-1 expression, our findings indicate that sequential therapy with axitinib rechallenge after nivolumab resistance is reasonable for treating mRCC.
Response:
We appreciate your comments and also have adjusted our manuscript accordingly. The revised manuscript should be more comprehensive.

Round 2

Reviewer 1 Report

Thank you for responding to comments. However, the authors have not been able to resolve deficiencies in biomarker analysis results. To avoid misleading the reader, further careful consideration is needed for the paper to be accepted.